# Molecular Insights into the Reproductive Patterns and Genetic Structure of Wheat Stripe Rust in Ili, Xinjiang

**DOI:** 10.3390/ijms252212357

**Published:** 2024-11-18

**Authors:** Hanlin Lai, Yue Li, Feifei Deng, Hong Yang, Jin Li, Jianghua Chen, Jingjing Sun, Guangkuo Li, W. G. Dilantha Fernando, Haifeng Gao

**Affiliations:** 1Institute of Plant Protection, Xinjiang Academy of Agricultural Sciences/Key Laboratory of Integrated Pest Management on Crop in Northwestern Oasis, Ministry of Agriculture and Rural Affairs, Urumqi 830091, China; lhanlin6218@163.com (H.L.); xjzbskg@163.com (F.D.); yh2201832022@163.com (H.Y.); jinli@xaas.ac.cn (J.L.); jianghuachen2024@126.com (J.C.); sunjingj2022@163.com (J.S.); lgk0808@163.com (G.L.); 2College of Life Science, Xinjiang Agricultural University, Urumqi 830001, China; liyue6905@126.com; 3Department of Plant Sciences, University of Manitoba, Winnipeg, MB R3T 2N2, Canada

**Keywords:** wheat stripe rust, simple sequence repeats, genetic structure, sexual reproductive

## Abstract

Wheat stripe rust, caused by *Puccinia striiformis* f. sp. *tritici* (*Pst*), is a globally significant fungal disease that seriously threatens wheat yield, particularly in China. This study investigates the genetic structure and reproductive patterns of *Pst* populations in Ili, Xinjiang, using 12 pairs of Simple Sequence Repeat (SSR) molecular markers. Analyses of 79 *Pst* isolates from either spring or winter wheat areas in Ili revealed three primary genetic clusters, indicating notable differences between populations associated with spring and winter wheat. The STRUCTURE results, complemented by UPGMA and PCoA analyses, highlight significant genetic diversity within these populations, with evidence of genetic recombination and sexual reproduction in certain areas. *Pst* populations in Ili exhibit a mixed mode of reproduction, predominantly sexual in Qapqal and Xinyuan D and primarily asexual within the spring wheat populations. The gene flow analysis underscores extensive inter-population communication, which facilitates the spread and adaptation of the pathogen across diverse wheat-growing environments. This study marks the first documentation of sexual reproduction in *Pst* within Xinjiang, providing new insights into its spread and genetic variation. These findings suggest that sexual reproduction may play a role in the regional adaptation and evolution of *Pst*, impacting future management strategies for wheat stripe rust in Xinjiang and potentially in broader Central Asian contexts.

## 1. Introduction

Wheat stripe rust, caused by *Puccinia striiformis* f. sp. *tritici* (*Pst*), is a globally significant fungal disease that seriously threatens wheat yield [1]. China is among the countries most severely impacted by stripe rust epidemics [2]. Since the 1940s, this disease has persistently affected wheat production in China annually and caused significant losses. By 2020, China had experienced eight major pandemics of wheat stripe rust, affecting an average area of 6.167 million hectares [3,4]. Consequently, wheat stripe rust has been listed as a priority for crop pest control within China. Pharmaceutical intervention and the selection of stripe rust-resistant wheat varieties are the primary strategies for preventing wheat stripe rust. The development and deployment of resistant varieties are currently recognized as the most effective, environmentally friendly, and cost-efficient approaches [5]. However, the resistance of these varieties often diminishes after 3–5 years in production due to the continuous emergence of new pathogen races of *Pst* [6]. This evolution is primarily driven by mutation, heterokaryosis and sexual reproduction, which enable *Pst* to produce new pathogen races [7].

Sexual reproduction is crucial in generating genetic diversity and fostering the emergence of new virulent races of pathogens [8]. For a long time, the wheat stripe rust pathogen was believed to reproduce solely asexually until Jin et al. [9] demonstrated that barberry (*Berberis* spp.) acts as the alternate host for the pathogen. This treatment was established by collecting samples of stripe rust pathogens from barberry in natural environments and subsequently inoculating them on wheat. In 2013, Zhao et al. [10] confirmed that wheat stripe rust pathogen can also reproduce sexually in natural environments. Sexual reproduction is a critical mechanism for generating new races of the stripe rust pathogen. However, traditional methods of identifying virulence patterns are inadequate for determining the potential for sexual reproduction in *Pst* populations. Instead, analyzing genetic structure and diversity of *Pst* populations through molecular marker methods enables the rapid monitoring of genetic recombination and inferring reproductive patterns within epidemic regions.

Sexual reproduction is considered to accelerate the adaptation of pathogens to their hosts by producing genetic diversity [11]. Sexual reproduction of *Pst* has been found extensively in China [10,12], unlike in other major wheat-producing areas such as Europe [13] and the United States [14], where it has not been observed. The genetic diversity of *Pst* in China is higher than that in both Europe and the Americas. Studies on the global genetic structure of *Pst* showed that the Himalayan region exhibits the highest genetic diversity, suggesting that it is the center of *Pst* origin [15]. Gansu Longnan and Northwest Sichuan are key regions where *Pst* exhibits significant seasonal variations during the summer in China [7]. These areas are crucial for providing sources of *Pst* to the eastern region of China and are notable for the initial discovery of new *Pst* races [7]. This significant genetic variability is most likely attributed to the sexual reproduction of *Pst* in this region. In addition, genetic recombination has also been observed in some populations in Qinghai of China [16]. The identification of *Berberis* species in Tibet revealed that all six surveyed species of barberry were alternate hosts of *Pst* [17]. These hosts are important for the epidemiology of wheat stripe rust and contribute to the genetic variability of *Pst*.

Ili, Xinjiang, a region with a high incidence of wheat stripe rust, provides climatic conditions that support the summer and winter survival of *Pst*. The prevalent distribution of both red-fruited and black-fruited barberry in this area indicates a potential for sexual reproduction in *Pst*. Prior studies in this region have primarily focused on identifying stripe rust races [18,19], with only a limited examination of genetic structures and no reported evidence of sexual reproduction among *Pst* populations. In our study, we utilized simple sequence repeat (SSR) markers to analyze the genetic structure and potential recombination within *Pst* populations in Ili. Our goal is to identify differences in genetic backgrounds across this region and to explore the occurrence of sexual reproduction.

## 2. Results

### 2.1. SSR Cumulative Genotypes

A gene accumulation curve analysis can effectively determine whether the SSR primers used in this experiment are adequate for assessing genetic diversity and genetic structure. In this study, 12 pairs of polymorphism-rich SSR primers were used, identifying a total of 36 alleles (Appendix A). The genotypic accumulation curve shows (Figure 1) that the genotypic richness reached a plateau when 11 pairs of primers were used, indicating that the 12 loci provided sufficient resolution for the genetic analysis in this research.

### 2.2. Genetic Structure Analysis

Genetic structure analysis of 79 *Pst* isolates based on Bayesian theory effectively distinguished the genetic background among different populations. A STRUCTURE analysis identified (Figure 2A) that K = 3 was the optimal value for sub-populations. All isolates were classified into three groups, including red (G1), green (G2), and blue (G3). G1, containing the largest number of isolates, encompasses isolates of *Pst* from all sampling regions of Ili, indicating a widespread genetic similarity across these areas. G2, which constitutes the next largest group, primarily includes isolates from regions such as Qapqal and Xinyuan D. The smallest group, G3, is predominantly composed of isolates from the spring wheat regions of Xinyuan C and Zhaosu, along with some from Gongliu. The results of the STRUCTURE analysis suggest that different *Pst* populations were more similar to each other in terms of the genetic background, and a distinct genetic segregation between spring and winter wheat populations. PCoA and DAPC analyses were conducted to corroborate the findings from the STRUCTURE analysis. PCoA results grouped all isolates into two clusters: one cluster on the left side primarily included spring wheat isolates from Xinyuan C and Zhaosu, whereas the cluster on the right side comprised isolates from all strains in Ili (Figure 2). This separation indicates significant genetic communication among the *Pst* populations, with notable distinctions occurring only in a few spring wheat strains. Similarly, a DAPC analysis further refined these distinctions, effectively differentiating the Xinyuan C group from other groups and corroborating the STRUCTURE results (Figure 3). These analyses collectively highlight the complex genetic landscape of *Pst* across different wheat crops and regions.

### 2.3. Genetic Diversity Analysis

Using 12 pairs of SSR primers, 79 *Pst* isolates from both spring and winter wheat in Ili were genotyped, revealing a total of 62 unique genotypes (Figure 4). Of these, 45 genotypes were detected in spring wheat, and 17 were found in winter wheat. The Gongliu population exhibited the highest number of unique genotypes. Notably, all the genotypes were shared among the Qapqal, Nilka, Zhaosu, and Tekesi populations, highlighting significant genetic overlap. Five of the sixty-two genotypes were shared by all populations. The most prevalent, MLG61, was found within the Gongliu, Xinyuan D, and Tekes populations. There were also shared genotypes between Gongliu and Nilka and between Zhaosu and Qapqal. These findings suggest substantial genetic exchange between the Gongliu population and other regional groups, providing insights into the pathways for stripe rust migration within the region. The genetic diversity index was calculated based on the 62 genotypes of *Pst* populations in Ili. The overall genetic diversity index (H) was 3.96, indicating high genotypic diversity across the sampled populations. Diversity indices among these populations varied, ranging from 1.30 to 2.87. The lowest diversity index was observed in Tekes (1.39), and the highest in Gongliu (2.87) (Table 1). This variation in genetic diversity suggests differing levels of genetic variability across populations, which may be influenced by factors such as host variety and local climatic conditions.

### 2.4. Genetic Recombination Analysis

Linkage disequilibrium analysis is an effective method to dissect reproduction patterns within populations. In this study, the hypothesis of sexual reproduction within *Pst* populations was verified using the standardized index of association (rbarD). The observed rbarD values ranged from −0.19 in the Tekes population to 0.24 in the Xinyuan C population, suggesting the possibility of linkage equilibrium and the occurrence of sexual reproduction in Qapqal, Tekes, and Xinyuan D populations (*p* > 0.05) (Figure 5). The correlation analysis among molecular markers revealed a low linkage equilibrium between CPS27 and CPS34; however, a strong linkage disequilibrium was noted across the remaining 11 loci (Figure 6). In summary, we hypothesize that the *Pst* populations in Ili exhibit a mixed mode of reproduction, predominantly sexual in Qapqal and Xinyuan D and primarily asexual within the spring wheat populations.

### 2.5. Gene Flow Between Pst Populations

Based on the genetic differentiation coefficients calculated from the seven populations of *Pst* in Ili, the gene flow between the populations was estimated according to the formula Nm = 0.25(1 − Fst)/Fst. The computed gene flow values ranged from 1.562 to 1097.502, with the highest gene flow between Xinyuan C and Zhaosu (1097.502), followed by the gene flow between Qapqal and Tekes (48.987), and the lowest gene flow between Xinyuan C and Nilka (1.561) (Table 2). Additionally, gene flow values between Zhaosu and all other populations exceeded four. The extensive gene flow indicates significant inter-population communication within the *Pst* populations in the Ili region, suggesting active migration among these populations.

### 2.6. Clustering Analysis of Pst Isolates in Ili

The genetic relationships among *Pst* isolates were assessed using UPGMA based on Nei’s genetic distance. This analysis revealed that the isolates could be classified into four molecular groups (MGs) (Figure 7A). MG1 exclusively contained isolates from Zhaosu and Xinyuan C. Similarly, MG3 consisted solely of isolates from Nilka and Gongliu. In contrast, MG2 and MG4 contained isolates from all the *Pst* groups in Ili. Further analysis of the genetic relationships among these populations was conducted using the UPGMA method (Figure 7B). The closest genetic relationship was observed between the Qapqal and Xinyuan D populations, while a close relationship was also noted between the Zhaosu and Xinyuan C populations, which aligns with the molecular clustering findings.

## 3. Discussion

In this study, we assessed the genetic diversity and population structure of *Pst* populations from spring and winter wheat in Ili, Xinjiang, using 12 pairs of SSR molecular markers. The investigation revealed a rich genetic diversity with a total of 62 unique genotypes identified among the 79 *Pst* isolates analyzed. Notably, three primary genetic clusters were classified, indicating significant differences between the *Pst* populations associated with spring and winter wheat cultivars. The gene flow analysis underscores extensive inter-population communication, which facilitates the spread and adaptation of the pathogen across diverse wheat-growing environments. Furthermore, the detection of genetic recombination and sexual reproduction within these populations marks a critical discovery, providing the first evidence of such reproductive behavior in *Pst* within Xinjiang.

Ili, characterized by both its spring and winter wheat cultivation zones, exhibits significant differences in maturity periods and extended growth cycles, which facilitate the cyclic spread of stripe rust [20]. Previous research identifying stripe rust races in these regions confirmed the presence of identical races on both spring and winter wheat, suggesting that *Pst* can migrate from winter to spring wheat crops [18]. In our study, we used molecular markers to analyze the genetic diversity and structure of *Pst* in Ili. We found that genetic group G3 was mainly distributed in spring wheat-growing areas such as Xinyuan C, Zhaosu, and Gongliu, whereas groups G1 and G2 were distributed in both spring and winter wheat populations. This distribution pattern likely stems from variations in planting practices, which contribute to some differences in the stripe rust populations. However, most individual populations belong to the same genetic group, supporting the hypothesis that *Pst* can be transmitted from winter to spring wheat, as demonstrated through genetic structure analysis.

Sexual reproduction is a crucial mechanism for generating genetic variation in wheat stripe rust. Extensive research, including a study by Jiang et al., who analyzed 2103 isolates across China using SSR molecular markers, has documented significant sexual reproduction in populations from the southwest and northwest regions [21]. Further genetic analysis of *Pst* populations in regions such as Shaanxi and Gansu revealed a linkage equilibrium, suggesting cellular or sexual recombination in areas including Qinzhou, Qincheng, Beidao, Maiji, Longxian, and Qianyang [22]. Indeed, patterns of sexual reproduction have been observed in several northwestern regions like Gansu [22], Shaanxi [12], and Qinghai [16]. However, despite its similar geographic location, there have been no reports of sexual reproduction in *Pst* from Xinjiang. The widespread presence of barberry in Xinjiang, including red-fruited and black-fruited species in Ili, provides ideal conditions for the sexual cycle of *Pst*, indicating a potential for sexual reproduction in this region. Consistent with this finding, our study detected a linkage equilibrium in the populations of Xinyuan D, Qapqal, and Tekes. These findings, which propose the potential occurrence of sexual reproduction of *Pst* in Xinjiang Ili based on genetic structure for the first time, offer a novel approach to the prevention and control of wheat stripe rust in the region.

## 4. Materials and Methods

### 4.1. Sample Collection

Sample collection was conducted in the winter (June 2021) and spring (July 2021) wheat-growing areas from 41 sites across six counties in Ili (Appendix A). Five wheat leaves infected with stripe rust were collected from each site. To prevent cross-contamination, each leaf was individually placed in a sulfate paper bag. The geographic location and elevation of each sampling site were recorded. Based on the geographic diversity and timing of the collections, all isolates were categorized into seven distinct populations of *Pst*. Specifically, isolates from Qapqal County were designated as ‘Qapqal’, those from Gongliu County as ‘Gongliu’, from Nilka County as ‘Nilka’, from the spring wheat field in Xinyuan County as ‘Xinyuan C’, from the winter wheat field in Xinyuan County as ‘Xinyuan D’, from Zhaosu County as ‘Zhaosu’, and from Tekes County as ‘Tekes’.

### 4.2. Sample Isolation and Propagation

The collected leaf samples were rinsed with water to remove surface impurities, then cut into small pieces and placed on moistened absorbent paper. The surface of the leaves was sprayed with water and subsequently incubated at 11 °C for 12 h to ensure the complete germination of summer spores. Using the tip of a needle, a single summer spore pile was isolated and transferred to the adaxial surface of a leaf on a wheat seedling. To ensure that each plant was infected by only one spore pile, each inoculated pot was enclosed with a plastic dome to maintain high humidity and a low temperature and returned to an 11 °C environment for 24 h to facilitate successful infection by the stripe rust fungus. Post-inoculation, the dark-treated wheat was exposed to a light regime of 12 h of light and 12 h of dark at 11 °C for cultivation. After two weeks, the summer spores proliferated on the wheat leaves. Next, all collected isolates were processed into spore suspensions with concentrations ranging from 2 to 5 mg/mL using 3M Novec TM 7100 e-fluoride. Subsequently, 5 µL of the mixed spore suspension was aspirated with a pipette gun and inoculated onto the adaxial surface of the leaves of the susceptible variety, Minh-Hsien 169. This step was crucial for amplifying all isolates to gather a sufficient quantity of urediniospores for subsequent experiments.

### 4.3. DNA Extraction and Genotyping of Pst

For DNA extraction, 10 mg of *Pst* were used following a modified CTAB method [23]. Genotyping was conducted using a set of 12 pairs of SSR primers [20,21,22,23,24] listed in Table 3. All primers were synthesized by Shanghai Sangong Biotech Co., Shanghai, China, and the corresponding fluorescent markers (FAM, ROX, TAMRA, or HEX) were added. The PCR was performed using a 25 µL system (including 12.5 µL of 2× Easy Taq PCR SuperMix, 1 µL of template (100 ng/µL), 1 µL of forward primer and 1 µL of reverse primer, and 9.5 µL of ddH_2_O), and the reaction program consisted of 5 min at 94 °C, for 30 s at 94 °C, 30 s at 55–60 °C, for 30 s at 72 °C, for 30 s, 35 cycles, 72 °C for 10 min, and a hold at 10 °C indefinitely. The PCR products were then loaded onto a 96-well plate and analyzed with a 3500 xl Genetic Analyzer (Applied Biosystems, Waltham, MA, USA) for PCR fragment length detection. Data analysis was performed using GeneMaker software v2.7.0 (SoftGenetics, LLC, State College, PA, USA).

### 4.4. Data Analysis

The analysis of the genetic structure was conducted by STRUCTURE 2.3 software [28], with the parameter K varying from 2 to 5. The optimal values were determined by STRUCTURE SELECTOR (https://lmme.ac.cn/StructureSelector/ (accessed on 5 July 2024)) after 15 repeated runs. GenAlEx 6.5 [29] was used to perform the two-dimensional principal coordinate analysis (PCoA) based on Nei’s genetic distance [30]. This software was also used to calculate the genetic divergence (Fst) between populations from different regions, and inter-population gene flow was estimated using the formula Nm = 0.25(1 − Fst)/Fst, as well as to assess genetic diversity within populations using Shannon’s Information Index (I). The clustering of isolates was performed based on the UPGMA method with POWERMARKER 3.25 software [31]. For graphical representation and enhancement, MEGA 11.0.8 software and ITOL online website (https://itol.embl.de/ (accessed on 5 July 2024)) were employed. The data format was exported by GenAlEx 6.5 [29], and statistics of genotypic combinations of 12 SSR markers (multilocus genotypes, MLGs) for each population, association indices among SSR markers (correlation standard index, rbarD), the Shannon–Wiener index of MLG diversity (H), the linkage disequilibrium, and shared multilocus genotypes among populations were performed using the poppr 2.20 package [32] in R. Furthermore, a discriminant analysis of principal components (DAPCs) based on multivariate discrimination was performed with the ‘DAPC’ function in the adegenet package of R [33]. The results were visualized using the ‘scatter.dapc’ function.

## 5. Conclusions

This study employed 12 SSR markers to explore the genetic structure and reproduction pattern of *Pst* populations from Xijaing Ili. The experimental findings confirm that the differences between *Pst* populations from spring and winter wheat areas and *Pst* can migrate from winter to adjacent spring wheat based on genetic structure analyses. Notably, this study documented the presence of sexual reproduction in *Pst* within Ili, marking the first such detection in this region following earlier confirmations in the southwest and northwest of China using molecular marker technology. Future research will extend this genetic analysis to encompass *Pst* populations throughout Xinjiang and Central Asian countries. This research will help clarify the source of *Pst* and its spread among populations in these regions.

## Figures and Tables

**Figure 1 ijms-25-12357-f001:**
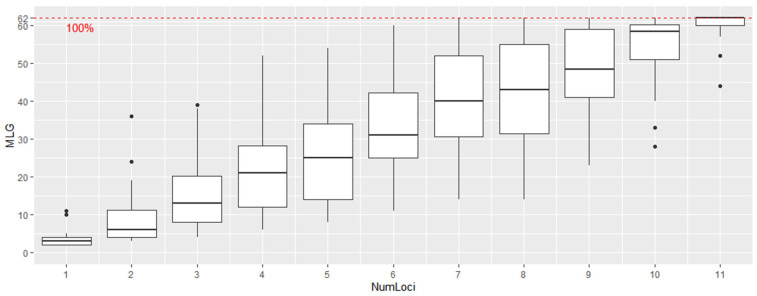
Genotype accumulation curve under the superposition of 12 loci. The *x*-axis represents the number of SSR loci. The *y*-axis (MLG) indicates the multilocus genotypes, showing genotypic richness. The maximum number of polygenic locus genotypes identified was 62, utilizing 11 SSR markers.

**Figure 2 ijms-25-12357-f002:**
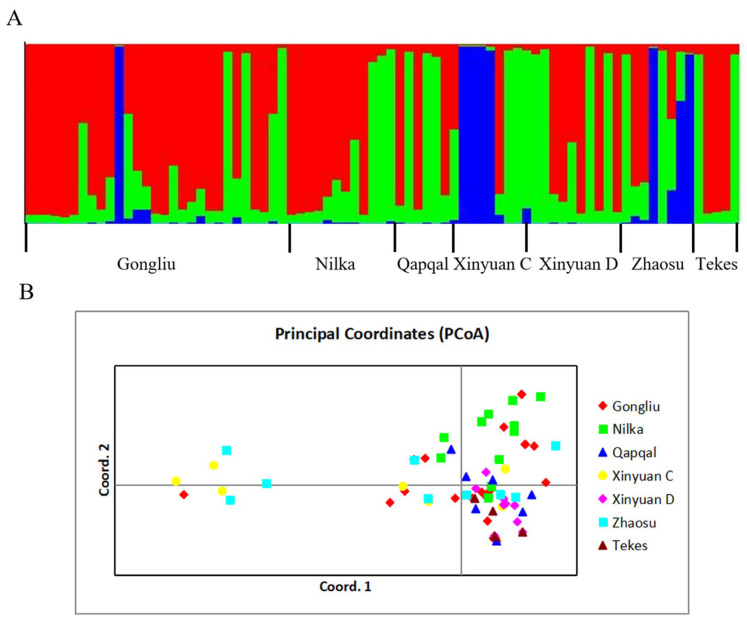
Structure analysis of *Puccinia striiformis* f. sp. *tritici* isolates from Ili: (**A**) STRUCTURE analysis of 79 *Pst* isolates in Ili, and (**B**) principal coordinate analysis (PCoA) of 79 *Pst* isolates in Ili.

**Figure 3 ijms-25-12357-f003:**
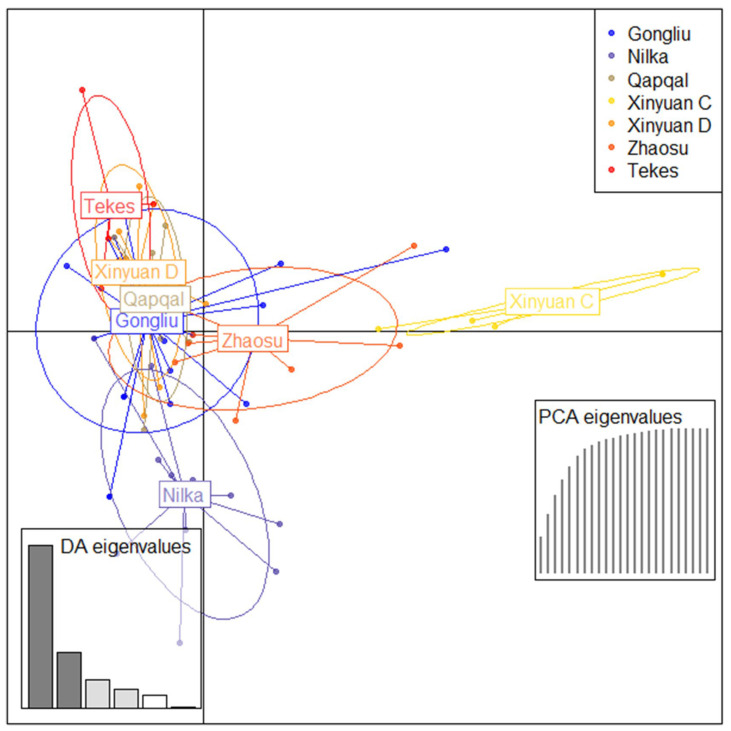
Discriminant analysis of principal components (DAPC) of 79 *Pst* isolates in Ili.

**Figure 4 ijms-25-12357-f004:**
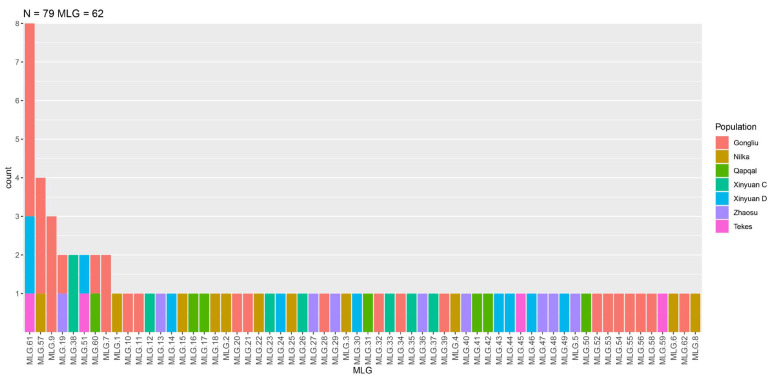
Genotypes of 79 *Pst* isolates in Ili. The chart categorizes these genotypes based on the population source. The *x*-axis displays individual MLGs, ranging from MLG1 to MLG62, highlighting the genotypic diversity within the region. The *y*-axis measures the count of each genotype within each population.

**Figure 5 ijms-25-12357-f005:**
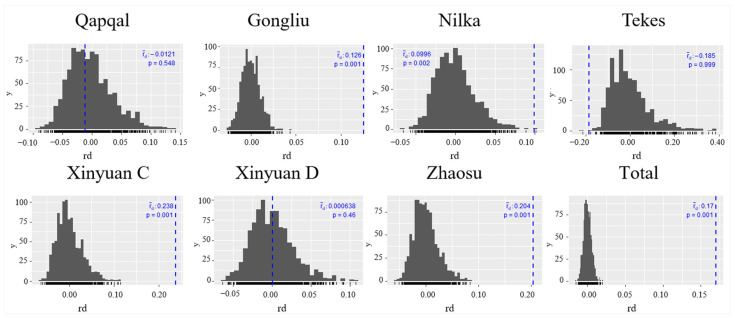
Standardized correlation coefficient diagram of linkage disequilibrium analysis of seven *Pst* subpopulations in Ili.

**Figure 6 ijms-25-12357-f006:**
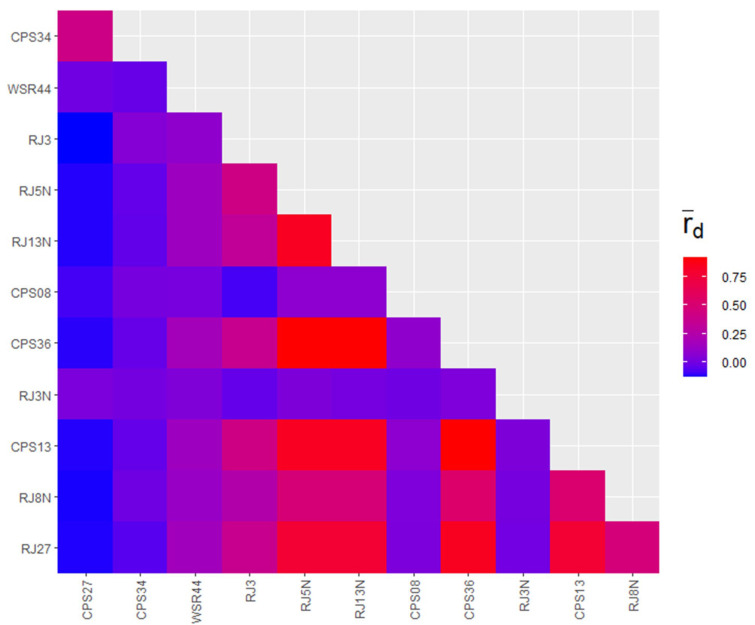
Linkage disequilibrium among 12 SSR markers within *Pst* populations. rd: correlation standard index.

**Figure 7 ijms-25-12357-f007:**
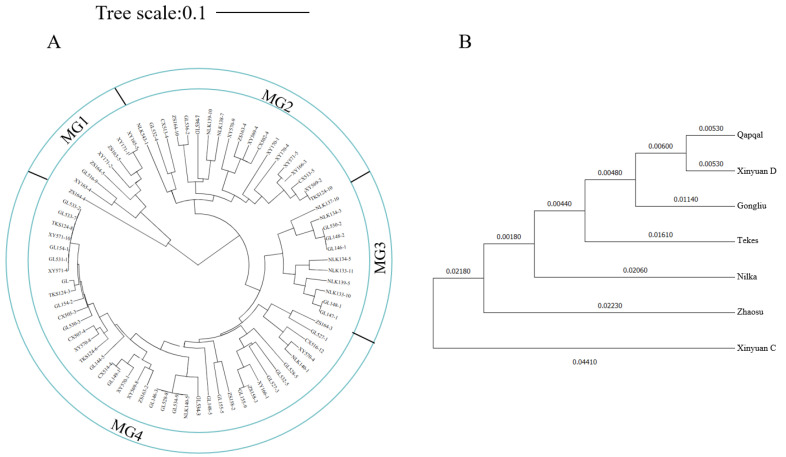
Molecular clustering and population clustering of each strain of *Pst* in Ili: (**A**) molecular clustering analysis using the UPGMA method based on Nei’s genetic distance between strains; and (**B**) UPGMA clustering based on Nei’s genetic distance between populations.

**Table 1 ijms-25-12357-t001:** Genetic diversity and multilocus linkage disequilibrium of *Pst* populations collected from spring and winter wheat cultivars in Ili.

Type of Host	Population	N	MLG	eMLG	H	rbarD	I
Spring wheat	Gongliu	30	21	8.56	2.87	0.13	0.729
Spring wheat	Nilka	11	11	10.00	2.40	0.09	0.729
Winter wheat	Qapqal	7	7	7.00	1.95	−0.01	0.719
Spring wheat	Xinyuan C	8	7	7.00	1.91	0.24	0.668
Winter wheat	Xinyuan D	10	9	9.00	2.16	0.00	0.694
Spring wheat	Zhaosu	9	9	9.00	2.20	0.20	0.795
Spring wheat	Tekes	4	4	4.00	1.39	−0.19	0.569
	Total	79	62	9.47	3.96	0.17	0.700

Note: N: number of samples from different populations; MLG: number of multilocus genotypes; eMLG: number of effective multilocus genotypes; H: Shannon–Wiener index of MLG diversity; rbarD: correlation standard index; and I: Shannon’s information index.

**Table 2 ijms-25-12357-t002:** Gene flow of *Pst* populations in Ili.

	Gongliu	Nilka	Qapqal	Xinyuan C	Xinyuan D	Zhaosu	Tekes
Gongliu	0.000						
Nilka	7.774	0.000					
Qapqal	-	6.501	0.000				
Xinyuan C	2.052	1.561	2.245	0.000			
Xinyuan D	27.076	3.158	-	1.612	0.000		
Zhaosu	9.548	5.526	35.080	1097.502	6.810	0.000	
Tekes	-	2.644	48.987	1.848	-	7.163	0.000

**Table 3 ijms-25-12357-t003:** Summary of SSR markers used in this study.

Locus	Repeat Motif	Primer Sequence (5′-3′)	Size	No. of Alleles	Reference
CPS08	(CAG)_14_	FAM-GATAAGAAACAAGGGACAGC	205–208	2	[24]
		CAGTGAACCCAATTACTCAG			
CPS13	(GAC)_6_	FAM-TCCAGGCAGTAAATCAGACGC	125–128	2	[24]
		ATCAGCAGGTGTAGCCCCATC			
CPS27	(TTC)_4_	TAMRA-GATGGGGAAAAGTAAGAAGT	222–225	2	[24]
		GGTGGGGGATGTAAGTATGTA			
CPS34	(TC)_9_	HEX-GTTGGCTACGAGTGGTCATC	105–113	6	[24]
		TAACACTACACAAAAGGGGTC			
CPS36	(CTCTAG)_3_	TAMRA-TCCAGGCAGTAAATCAGACGC	125–128	2	[24]
		ATCAGCAGGTGTAGCCCCATC			
RJ3	(TGG)8	ROX-GCAGCACTGGCAGGTGG	204–207	4	[25]
		GATGAATCAGGATGGCTCC			
RJ27	(TC)_10_	TAMRA-CGTCCCGACTAATCTGGTCC	230–242	2	[25]
		ATGAGTTAGTTTAGATCAGGTCGAC			
RJ3N	(CT)_9_	HEX-TGGTGGTGCTCCTCTAGTC	338–346	5	[26]
		AGGGGTCTTGTAAGATGCTC			
RJ5N	(CT)_8_	HEX-AACGGTCAACAGCACTCAC	225–227	2	[26]
		AGTTGGTCGCGTTTTGCTC			
RJ8N	(GAT)_8_	FAM-ACTGGGCAGACTGGTCAAC	304–308	3	[26]
		TCGTTTCCCTCCAGATGGC			
RJ13N	(ACG)_6_	ROX-TTAGCTCAGCCGGTTCCTC	149–152	2	[26]
		CAGGTGTAGCCCCATCTCC			
WSR44	(GT)_6_	HEX-AGGCCCCAGGAACACAAAAA	189–192	4	[27]
		TCACACACGCTCCACAGTAC			

## Data Availability

Data are contained within this article and Appendix A.

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
