# Peer review of "Molecular Insights into the Reproductive Patterns and Genetic Structure of Wheat Stripe Rust in Ili, Xinjiang"

_ijms, 2024, doi:10.3390/ijms252212357_

Round 1

Reviewer 1 Report

Comments and Suggestions for Authors

1. Please provide more details on how the indices you used in the article were calculated: rbarD, MLG, еMLG, H, I. 

2. There is no information on how you calculate linkage disequilibrium. Give more information.

Author Response

Dear Editors and Reviewers:

Thank you for your letter and for the reviewers’ comments concerning our manuscript entitled “Molecular Insights into the Reproductive Patterns and Genetic Structure of Wheat Stripe Rust in Ili, Xinjiang”. Those comments are all valuable and extremely helpful for revising and improving our paper, as well as providing important guiding significance to our research. We have carefully evaluated and discussed the contents of comments and made correction which we hope would meet the demand of publication.

Comments: 1. Please provide more details on how the indices you used in the article were calculated: rbarD, MLG, еMLG, H, I.

  1. There is no information on how you calculate linkage disequilibrium. Give more information.

Response: I have followed the modifications and described the calculation of rbarD, MLG, еMLG, H, I and linkage disequilibrium in Materials and Methods, in lines 299-302.

Reviewer 2 Report

Comments and Suggestions for Authors

Dear Authors,

Reviewer comments ijms-3289364

The manuscript entitled „Molecular insights into the reproductive patterns and geentic structure of wheat stripe rust in Ili, Xinjiang“ represents a useful study aimed at an investiagtion and characterization of wheat stripe rust Puccinia striiformis Pst populations sample dat 7 locations in Ili, Xinjiang, China. I can recommend the manuscript for publication in International Journal of Molecular Sciences. I have only a few minor comments on the present manuscript version which are provided below.

1/ Terminology: The terms related to genetic characterization of Pst strains such as ůmultilocus genotypes“ and „genotypic richness“ have to be briefly explained in Introduction for unfamiliar readers.

2/ Figure 3 legend should be modified as follows. „DAPC analysis (full name has to be given) of 79 Pst isolates sample dat 7 locations in Ili, Xinjiang.“

3/ In Figure 6 legend, the symbol rd and the scale have to be explained in Figure 6 legend.

4/ In Figure 7A providing the results of molecular clustering, any statistical evaluation has to be added to the phylogenetic tree since it is otherwise just a scheme; i.e., either numbers at nodes providing bootstrap values per 1,000 replicates or a scale bar providing a genetic distance have to eb added to Figure 7A.

5/ In Materials and methods, the type of cultivation box used and the manufacturer has to be specified. Line 274, the manufacturer or any other reference has to be added to the software „Genemaker software v2.7.0.“

6/ Formal comments on the text related to English language and style:

Introduction, line 52: Add the word „treatment“ following „this“ and add a comma both prior to and after the word „subsequently“ in the statement: „This treatment was established by collecting samples of stripe rust pathogens from barberry in natural environments and, subsequently, inoculating them on wheat.“

Figure 1 legend should be more explanatory for teh unfamiliar reader.

Discussion, line 219: Modify „which“ to „who“ in the statement: „Extensive research including a study by Jiang et al. [21] who analyzed 2,103 isolates across China using SSR markers has documented significant sexual reproduction in populations from southwest and northwest regions.“

Discussion, line 230: Add the word „finding“ following the word „this“ in the statement: „Consistent with this finding, our study detecetd linkage equilibrium in the populations…“

Materials and methods, line 279: Correct the typing error in the software name „STRUCTURE HARVESTER“ (not „HARVESTE“).

Conclusions, line 301: Add „research“ following „this“ in the statement. „This research will help clarify the source of Pst…“

Final recommendation: Accept after a minor revision.

Author Response

Dear Editors and Reviewers,

We highly appreciate the comments on our manuscript entitled “Molecular Insights into the Reproductive Patterns and Genetic Structure of Wheat Stripe Rust in Ili, Xinjiang”. We have taken the comments on board to improve and clarify the manuscript. Please find below a detailed point by-point response to all comments.We appreciate the reviewer's comments. I have revised the manuscript as you have already done:

Comments1: Terminology: The terms related to genetic characterization of Pst strains such as “multilocus genotypes” and “genotypic richness” have to be briefly explained in Introduction for unfamiliar readers.

Response1: I have briefly explained the terms “multilocus genotypes” and “genotypic richness” in lines 295-302 as per the amendment.

Comments2: Figure 3 legend should be modified as follows. “DAPC analysis (full name has to be given) of 79 Pst isolates sample dat 7 locations in Ili, Xinjiang.”

Response2: The following changes have been made as requested: “Discriminant analysis of principal components (DAPC) of 79 Pst isolates in Ili.” in line 128.

Comments3: In Figure 6 legend, the symbol rd and the scale have to be explained in Figure 6 legend.

Response3: The rd has been annotated in the figure notes to Figure 6 as required by the modifications.

Comments4: In Figure 7A providing the results of molecular clustering, any statistical evaluation has to be added to the phylogenetic tree since it is otherwise just a scheme; i.e., either numbers at nodes providing bootstrap values per 1,000 replicates or a scale bar providing a genetic distance have to eb added to Figure 7A.

Response4: I've added “Tree scale: 0.1” to Figure 7A.

Comments5: In Materials and methods, the type of cultivation box used and the manufacturer has to be specified. Line 274, the manufacturer or any other reference has to be added to the software „Genemaker software v2.7.0.“

Response5: Added “Genemaker software v2.7.0” to line 284 to add the manufacturer and origin of the software to the manuscript. Revise as follows: “GeneMaker software v2.7.0 (SoftGenetics, LLC, State College, PA, USA).”

Comments6: Formal comments on the text related to English language and style:

Introduction, line 52: Add the word „treatment“ following „this“ and add a comma both prior to and after the word „subsequently“ in the statement: „This treatment was established by collecting samples of stripe rust pathogens from barberry in natural environments and, subsequently, inoculating them on wheat.“

Figure 1 legend should be more explanatory for teh unfamiliar reader.

Discussion, line 219: Modify „which“ to „who“ in the statement: „Extensive research including a study by Jiang et al. [21] who analyzed 2,103 isolates across China using SSR markers has documented significant sexual reproduction in populations from southwest and northwest regions.“

Discussion, line 230: Add the word „finding“ following the word „this“ in the statement: „Consistent with this finding, our study detecetd linkage equilibrium in the populations…“

Materials and methods, line 279: Correct the typing error in the software name „STRUCTURE HARVESTER“ (not „HARVESTE“).

Conclusions, line 301: Add „research“ following „this“ in the statement. „This research will help clarify the source of Pst…“

Response6: Linguistic changes were made in line 52, 228, 239, 290 and 316, respectively, in accordance with the modifications. Figure 1 was modified as follows “Figure 1. Genotype accumulation curve under the superposition of 12 loci. The x-axis represents the number of SSR loci. The y-axis (MLG) indicates the multilocus genotypes, showing the genotypic richness. axis (MLG) indicates the multilocus genotypes, showing the genotypic richness. The maximum number of polygenic locus genotypes detected was 62 genotypes when 11 SSR markers were detected. The maximum number of polygenic locus genotypes detected was 62 genotypes when 11 SSR markers were used.”